# The ribosomal P-stalk couples amino acid starvation to GCN2 activation in mammalian cells

Heather P Harding[1]*, Adriana Ordonez[1], Felicity Allen[2], Leopold Parts[2], Alison J Inglis[3†], Roger L Williams[3], David Ron[1]*

[1]Cambridge Institute for Medical Research, University of Cambridge, Cambridge, United Kingdom; [2]Wellcome Trust Sanger Institute, Wellcome Genome Campus, Hinxton, United Kingdom; [3]Medical Research Council Laboratory of Molecular Biology, Cambridge, United Kingdom

**Abstract** The eukaryotic translation initiation factor 2α (eIF2α) kinase GCN2 is activated by amino acid starvation to elicit a rectifying physiological program known as the Integrated Stress Response (ISR). A role for uncharged tRNAs as activating ligands of yeast GCN2 is supported experimentally. However, mouse GCN2 activation has recently been observed in circumstances associated with ribosome stalling with no global increase in uncharged tRNAs. We report on a mammalian CHO cell-based CRISPR-Cas9 mutagenesis screen for genes that contribute to ISR activation by amino acid starvation. Disruption of genes encoding components of the ribosome P-stalk, uL10 and P1, selectively attenuated GCN2-mediated ISR activation by amino acid starvation or interference with tRNA charging without affecting the endoplasmic reticulum unfolded protein stress-induced ISR, mediated by the related eIF2α kinase PERK. Wildtype ribosomes isolated from CHO cells, but not those with P-stalk lesions, stimulated GCN2-dependent eIF2α phosphorylation in vitro. These observations support a model whereby lack of a cognate charged tRNA exposes a latent capacity of the ribosome P-stalk to activate GCN2 in cells and help explain the emerging link between ribosome stalling and ISR activation.

**\*For correspondence:**
hph23@cam.ac.uk (HPH);
dr360@medschl.cam.ac.uk (DR)

**Present address:** †California Institute of Technology, Pasadena, United States

## Introduction

Phosphorylation of translation initiation factor 2 on serine 51 of its alpha subunit (eIF2α) is a potent mechanism for translational regulation in eukaryotes. Phosphorylated eIF2 impedes the guanine nucleotide exchange activity of eIF2B thereby limiting the pool of active GTP-bound eIF2. The consequences to rates of translation initiation are mRNA-specific. By way of this direct effect on protein synthesis and its indirect consequences to the abundance of downstream effector proteins, levels of eIF2α phosphorylation modulate gene expression translationally and transcriptionally (*Hinnebusch, 2014*).

In animals four different kinases couple unrelated stress signals to eIF2α phosphorylation (eIF2α[P]). eIF2α[P] effectively integrates these into a stereotypical downstream response referred to as the Integrated Stress Response (ISR) (*Harding et al., 2003*). The ISR modulates biological processes ranging from the cell autonomous endoplasmic reticulum unfolded protein response to organismal immunity, memory and cognition (*Pakos-Zebrucka et al., 2016*; *Wek, 2018*). The four eIF2a kinases, GCN2, PERK, PKR and HRI, share a similar kinase effector domain, but diverge in the molecular mechanisms and nature of the upstream signals that regulate their kinase activities.

General Control Non-depressible 2 (GCN2) is the oldest eIF2α kinase, conserved in all known eukaryotes. It was discovered as the product of a gene required for yeast adaptation to starvation for any amino acid, as in its absence yeast were unable to mount a rectifying transcriptional General

**eLife digest** Often thought of as "workhorse" molecules, proteins take part in almost every structure and activity in a living cell. They are constructed from smaller building blocks called amino acids by molecular machines called ribosomes. Each cell needs a constant supply of amino acids to make new proteins. If cells are running low on amino acids, they can change their internal biochemistry to use amino acids more economically. GCN2 is one protein that helps activate these biochemical changes, but it was unclear how a shortage of amino acids could activate GCN2.

Earlier in 2019, researchers reported that, in a test tube at least, isolated ribosomes could themselves activate GCN2. They also identified a part of the ribosome called the P-stalk as playing an important role in the interaction. Now, Harding et al. – who include some of the researchers involved in the earlier study – explore the activation of GCN2 further, but this time based on experiments with mammalian cells.

First, a genetic screen was conducted to identify genes that if mutated specifically prevented the activation of GCN2 in cells that were starved of amino acids. This screen identified a few genes, several of which are involved in creating the P-stalk of the ribosome. By isolating the mutant ribosomes from these cells and studying them in the laboratory, Harding et al. then showed that these ribosomes are unable to activate GCN2.

These findings confirm that the P-stalk of the ribosome plays an essential role in activating GCN2 in response to a shortage of amino acids. They shed light on a fundamental biological system, and further work will undoubtedly seek to uncover the details of the process by which GCN2 is activated.

Control response (the yeast counterpart to animal cell ISR) (*Hinnebusch and Fink, 1983*). This amino acid starvation-induced, GCN2-dependent unicellular gene expression program has as its targets genes encoding transporters and biosynthetic enzymes that function to restore amino acid sufficiency, as well as tRNA synthetases that promote amino acid utilization as building blocks of proteins (*Hinnebusch, 2005*). These physiological features, in conjunction with the domain organization of the GCN2 protein (including an eIF2$\alpha$ kinase module and a module highly related to histidyl-tRNA synthetases, HisRS-like), suggested that uncharged tRNAs may serve as activating ligands of GCN2 as part of a feed-back mechanism that defends cellular pools of charged tRNAs (*Wek et al., 1989*; *Wek et al., 1990*). This model was supported by the observations that mutations in the HisRS-like module that interfere with uncharged tRNA-binding in vitro abolished GCN2 activity in yeast (*Wek et al., 1995*; *Zhu et al., 1996*) and by evidence that tRNA binding to the HisRS-like portion of GCN2 relieves an intramolecular repressive signal arising from its interaction with the kinase domain (*Dong et al., 2000*).

Parallel lines of enquiry indicate that GCN2 activity also relies on direct (*Ramirez et al., 1991*; *Zhu and Wek, 1998*) and indirect (*Marton et al., 1997*) interactions with ribosomes or ribosome-associated proteins (*Jiménez-Díaz et al., 2013*; *Inglis et al., 2019*). These latter observations suggest that GCN2 activation may not arise solely by binary interaction between GCN2 and uncharged tRNAs as activating ligands. Additionally, a recent genetic observation made in mice brought into question the singular role of excess uncharged tRNAs as GCN2 activating ligands and instead suggested that in some circumstances the interaction between GCN2 and ribosomes might take center stage. A strain of mice (C57BL/6J) that lacks an abundant neuron-specific isoacceptor arginyl-tRNA was noted to have higher levels of GCN2-dependent ISR activity in brain tissues compared with a reference strain that expressed normal levels of the neuron-specific arginyl-tRNA. ISR activity in arginyl-tRNA depleted C57BL/6J brain was not associated with globally elevated uncharged tRNAs, but was increased by a second mutation that compromises the cells ability to re-cycle stalled ribosomes (*Ishimura et al., 2016*). Together, these observations suggest a mechanism for activating mammalian GCN2 that emanates from a (stalled) ribosome-generated signal that arises when pools of charged tRNAs are limiting. Whilst a role for localized pools of uncharged tRNAs remains possible, the findings of Ishimura (et al. 2016) suggest that GCN2 activation may proceed independently of globally elevated levels of uncharged tRNAs.

Following up on these hints for the existence of additional aspects of mammalian GCN2 activation, we took advantage of recent developments in CRISPR-Cas9 technology to search for

mammalian genes whose compromise also enfeebles GCN2 activation. Our findings, pointing to a role for the ribosomal P-stalk in coupling an amino acid starvation-induced change in the ribosome to GCN2 activation, are described below.

## Results

### A genetic screen implicates the ribosomal P-stalk in ISR induction selectively by histidinol or amino acid deprivation

As a first step towards identifying genes implicated in GCN2-mediated ISR induction we confirmed the suitability of two reporter cell lines. In both HeLa and CHO cells inactivation of the GCN2-encoding *Eif2ak4* gene selectively abolished responsiveness of the ISR regulated CHOP::GFP reporter to the histidinyl-tRNA synthetase inhibitor, histidinol. In both cell lines, the CHOP::GFP reporter remained responsive to the glycosylation inhibitor tunicamycin, a toxin that activates the ISR orthogonally, through an ER stress inducible eIF2α kinase, PERK (*Figure 1A and B*). (*Harding et al., 1999*; *Harding et al., 2000*). Furthermore, GCN2 ablation did not affect the tunicamycin-responsive XBP1::mCherry reporter present in the CHO cells. The reporter cell lines were thus deemed suitable tools to search for additional components that may contribute to GCN2-dependent ISR activation.

The HeLa reporter cells were targeted with pooled lentivirus expressing guide RNAs targeting the entire human genome (*Sanjana et al., 2014*), treated with medium lacking lysine and arginine (-KR), and enriched for CHOP::GFP dull cells (that mimic the GCN2-ablation phenotype) using fluorescence-activated cell sorting (FACS) (*Figure 1C*). After two rounds of sorting, sequencing of the guides confirmed that those targeting the GCN2 encoding *Eif2ak4* were amongst the most highly enriched in the dull cells. Ontology cluster analysis also revealed enrichment for guides targeting the mRNA cap binding complex, other translation initiation factors, and ribosomal proteins, consistent with similar studies previously carried out in yeast (*Hinnebusch, 2005*).

Despite the enrichment for guides targeting genes plausibly implicated in amino acid starvation-mediated ISR activation, many of the HeLa cells selected for CHOP::GFP dullness had also lost the ability to respond to tunicamycin (not shown). Furthermore, it became evident that the clonogenic potential of stressed HeLa cells was poor, compared with CHO cells (*Figure 1—figure supplement 1A*). These features were deemed to compromise the prospect of enrichment for guides targeting genes of interest by further cycles of selection in HeLa cells. To circumvent this problem, we drew on the sequence information derived from the targeted HeLa cells to create a CHO-based CRISPR library focused on those genes enriched in the dull HeLa cells and expanded to other members of their gene families. The resulting 19,305-guide library targeting 3,222 CHO genes (~6 guides per gene) was used to mutagenize CHO cells. Flow cytometry-based sorting enriched for cells that were selectively compromised in CHOP::GFP expression in response to histidinol but retained significant responsiveness to tunicamycin (*Figure 1D*).

Sequencing of the integrated guides from genomic DNA isolated from the CHOP::GFP dull CHO cells confirmed enrichment of guides targeting *Eif2ak4* and a subset of genes encoding translation initiation factors and ribosomal proteins. Among the latter was *Rps10* (encoding eS10), previously implicated in GCN2 responsiveness to amino acid starvation in yeast (*Lee et al., 2015*) (*Figure 1—figure supplement 1B*). Guides targeting two other ribosomal genes were also conspicuously enriched in the dull CHO cells: *Rplp0* and *Rplp1*, encoding uL10 and P1, both components of the acidic ribosomal P-stalk, a heteropentameric structure that also includes P2 (guides directed to the P2-encoding *Rplp2* gene were also modestly enriched in the dull population) (*Figure 1E*).

The enriched *Rplp0* guides caught our interest, because they mapped to the region encoding the C-terminus of uL10, which constitutes the helical spine of the P-stalk protrusion from the ribosome surface and the linker connecting it to the ribosome core. Guides targeting the ribosome-embedded N-terminal portion of uL10 were strongly depleted from all pools of CHO cells (regardless of their ISR status) consistent with a role for this portion of the protein in cell fitness (*Figure 1E and F*). These findings hinted at a possible role for the P-stalk in ISR activation in response to histidinol. Given the proximity of the P-stalk to the ribosomal A site, we considered a role for the P-stalk in signaling event(s) triggered by lack of charged tRNAs or by ribosome stalling.

To follow up on the genotype-phenotype relationship suggested by the screen, we targeted CHO cells with the *Rplp0* and *Rplp1* guides found to be enriched in CHOP::GFP dull cells and

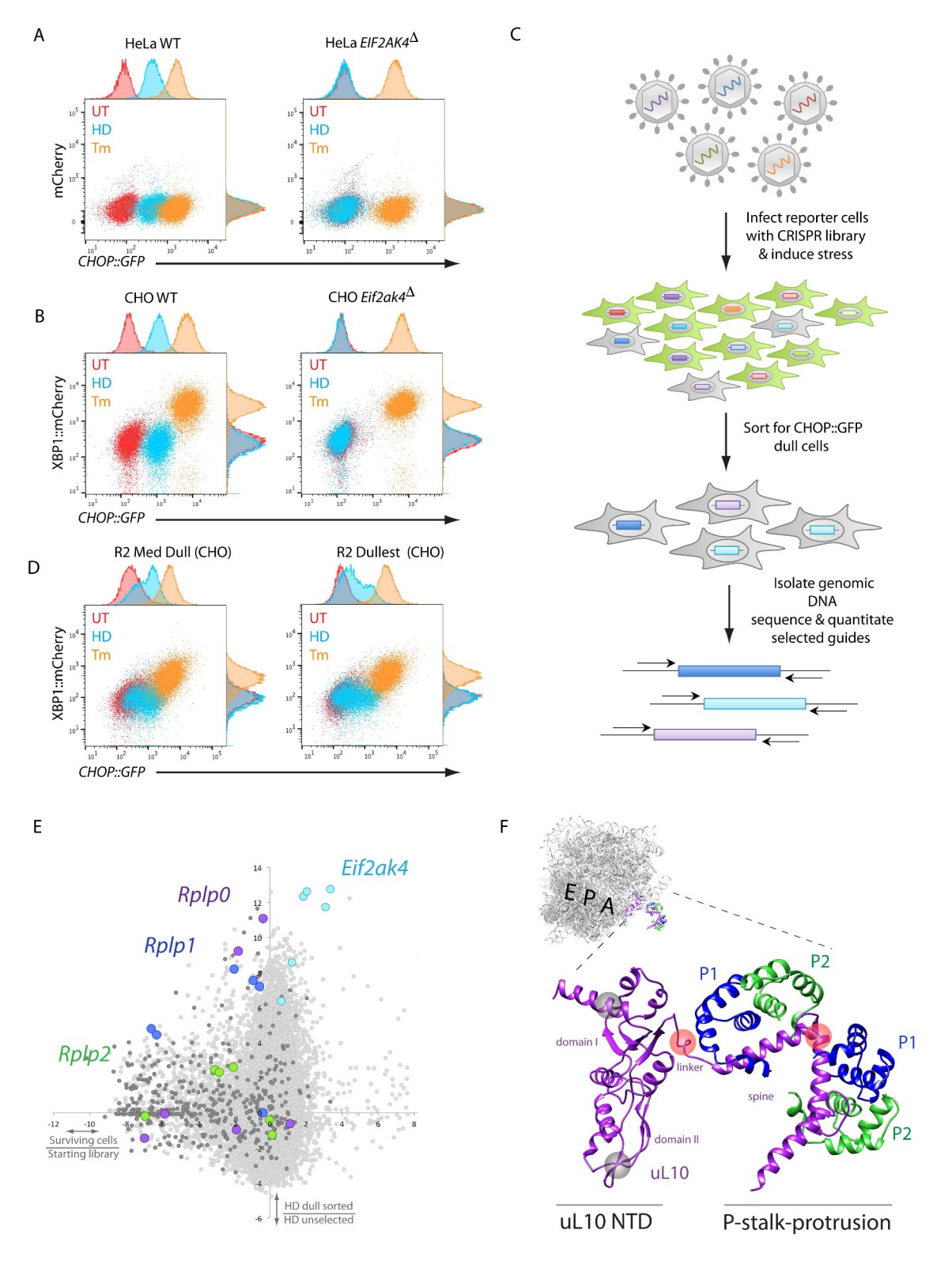

**Figure 1.** A CRISPR-Cas9-based genome-wide screen implicates the ribosomal P-stalk in ISR induction. (**A**) Overlay plot of the fluorescence signal from wildtype (WT) or GCN2-ablated *Eif2ak4*$^\Delta$ mutant HeLa cells with an ISR sensitive CHOP::GFP reporter (horizontal axis Ex: 488 nm/ Em530 ± 30 nm) and a constitutive mCherry reporter (vertical axis Ex 561/Em 610 ± 20 nm). The cells were untreated (UT, red), treated with histidinol (2 mM HeLa, 0.5 mM CHO, HD, blue) or tunicamycin (2 μg/ml, Tm, orange). Histograms of the signal in each channel are displayed on the axes. (**B**) Flow cytometry plot as in

*Figure 1 continued on next page*

*Figure 1 continued*

'A' but from wildtype or GCN2-ablated CHO cells with the CHOP::GFP reporter (horizontal axis) and an ER-stress sensitive XBP1::mCherry reporter (vertical axis). Color-coding as in A. (C) Schema used to enrich for cells with CRISPR-Cas9 induced genetic lesions that impair ISR activation (CHOP::GFP dull cells) and for identification of the guide RNA sequences they harbor. (D) Flow cytometry plot as in B but from pools of CRISPR-Cas9 mutagenized CHO cells following two rounds of FACS-based enrichment for histidinol-treated CHOP::GFP dull cells. 'R2 Med Dull' and 'R2 Dullest' refers to pools selected for medium or strong impairment of CHOP::GFP induction. (E) Plot of the mean $\log_2$ fold enrichment or depletion of guides. The vertical axis compares the histidinol-treated CHOP::GFP dull population to an unselected population of histidinol-treated CHO cells. The horizontal axis reports on the enrichment or depletion of guides from unselected cells compared to their abundance in the original library. Guides targeting *Eif2ak4*, *Rplp0*, *Rplp*1 and *Rplp*2 are color coded. All other ribosomal proteins are in dark grey. Note that only *Rplp0* guides that were not depleted from the unselected pool of transduced cells were enriched in the CHOP::GFP dull population. (F) Cartoon of the structure of the human ribosome with the position of the E, P, and A, sites highlighted and the P-stalk (based on PDB 4v6x) in close up. The ribosome associated N-terminal domain (NTD) of uL10 and the P-stalk protrusion are indicated. The unstructured acidic C-termini of uL10, P1, and P2, unresolved in PDB 4v6x, are not shown. The approximate positions on the protein corresponding to the site targeted by the *Rplp0* guides enriched in the CHOP::GFP dull cells or depleted from the unselected pool of transduced cells are indicated by the red and grey translucent spheres, respectively.

The online version of this article includes the following figure supplement(s) for figure 1:

**Figure supplement 1.** Poor clonogenic potential of stressed HeLa compared with CHO cells.

characterised genotypically-defined clones phenotypically. Histidinol induction of CHOP::GFP expression was conspicuously compromised in the targeted clones, whereas their responsiveness to ER stress-inducing agents was unaffected. A selective defect was also observed in *Rplp0* and *Rplp1* mutant cells in response to lysine and arginine depletion, albeit not to the level of GCN2 ablated cells (*Figure 2A and B*). The defect in the ISR brought about by targeting cells with the *Rplp0*-directed guide was rescued by restoring gene function, attained by re-targeting the mutant *Rplp0* locus with a homologous repair template to reestablish the coding region of the gene (whilst adding an epitope tag) (*Figure 2C and D*). These findings formally establish a correlation between CRISPR-Cas9-induced lesions in the P-stalk and a selective defect in the inducibility of the CHOP::GFP ISR reporter in response to inhibition of tRNA charging or to starvation for amino acids.

## P-stalk lesions affect the endogenous ISR

Attenuated translation initiation is a common feature of the ISR. It is readily detected by tracking the distribution of ribosomes in density gradients of cell lysates (*Nilsen et al., 1982*; *Harding et al., 2000*). Amino acid starvation of wildtype cells resulted in the expected redistribution of ribosomes from denser polysome fractions to lighter fractions and to free 40S and 60S subunits. This starvation-induced shift in the ribosome profile was lost in GCN2-ablated cells, but only partially attenuated even in the strongest single mutant lines (*Figure 3A and B*). This partial loss of function phenotype is consistent with the partial defect in CHOP::GFP induction and with the predicted structure of the mutant uL10, which retains the N-terminal portion of the P-stalk helical spine (*Figure 2A and B*).

In an effort to acquire a more penetrant lesion in the P-stalk, we targeted several *Rplp0* mutant clones with guides to *Rplp1*. Very few double mutant cells survived the procedure, but one such clone *Rplp0/Rplp*1[m29-132] (encoding a uL10 truncated at Y231 and a P1 protein lacking H17-D18 in helix 1) phenocopied GCN2 ablation both in terms of its polysome profile and activation of the CHOP::GFP reporter (*Figure 3A–3C*).

In wildtype cells amino-acid starvation led to a time dependent accumulation of activated hyper-phosphorylated GCN2, which was conspicuously lacking in the *Rplp0/Rplp*1[m29-132] cells. Note the retarded mobility of GCN2 from the starved wild type when resolved on a PhosTag gel, compared with GCN2 from mutant cells (*Figure 3D* and *Figure 3—figure supplement 1A*). Phosphatase treatment of the lysate, prior to gel loading, confirmed that the heterogenous mobility shift indeed reflected multiple phosphorylation events (*Figure 3E* and *Figure 3—figure supplement 1B*). Like the GCN2 ablated cells, the compound P-stalk *Rplp0/Rplp*1[m29-132] double mutant cells were also selectively defective in induction of the endogenous ISR markers CHOP and ATF4 in response to amino acid starvation but retained inducibility of the markers to ER stress (*Figure 3F*).

Immunoblotting of extracts from the wildtype and P-stalk mutant cells confirmed the truncation of uL10 (detected with an antisera to the N-terminal portion of the protein) and the loss of its C-terminus (revealed with a monoclonal antibody, 3BH5, reactive with a C-terminal peptide conserved in all three P-stalk proteins) in both the single *Rplp0*[m14] and the compound *Rplp0/Rplp*[m29-132] double

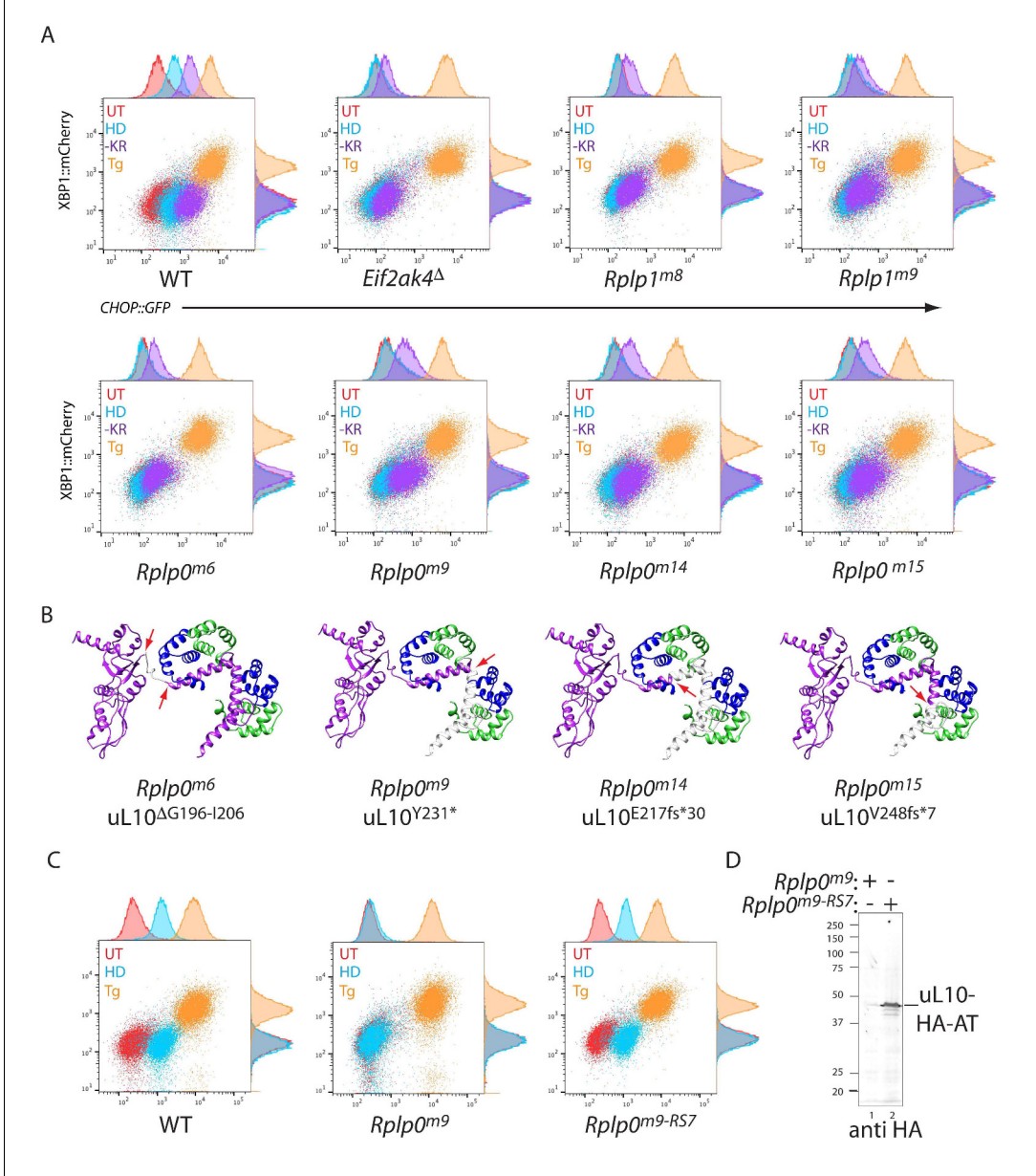

**Figure 2.** CRISPR-Cas9-based P-stalk lesions impair CHOP::GFP induction upon histidinol treatment or amino acid starvation. (**A**) Flow cytometry analysis of the ISR-inducible CHOP::GFP and the UPR inducible XBP1::mCherry reporters from untreated (UT red), histidinol-treated (0.5 mM, HD, blue), thapsigargin-treated (1 mM, Tg, orange), or cells starved for lysine and arginine (-KR, purple) all for 20 hr. Cells were targeted with guides to the indicated genes. (**B**) Depiction of the genetic lesion in the *Rplp0*-mutant clones whose flow cytometry profile is shown above. Shown is a ribbon diagram of the P-stalk (as in *Figure 1E*) Red arrows indicate the boundaries of the mutations and deleted regions are colored light grey. (**C**) Flow cytometry analysis of parental (wildtype, WT) CHO cells, the *Rplp0$^{m9}$* mutant and the mutant following restoration of the P-stalk by re-targeting the mutant *Rplp0* with a repair template encoding a wildtype allele with an HA epitope tag (*Rplp0$^{m9-RS7}$*) (**D**) Anti-HA immunoblot of cytoplasmic extracts from the parental *Rplp0$^{m9}$* and rescued *Rplp0$^{m9-RS7}$* cells.

mutant (*Figure 4A*). Coomassie-stained SDS-PAGE gels of ribosomes isolated from the mutant cells was conspicuous for the presence of a 34.2 kD protein (the predicted size of uL10) in the wildtype that was missing from both mutants (*Figure 4B*). Furthermore, ribosomes isolated from the mutant cells had diminished P1/P2 content (conspicuous in the *Rplp0/Rplp1$^{m29-132}$* double mutant but also

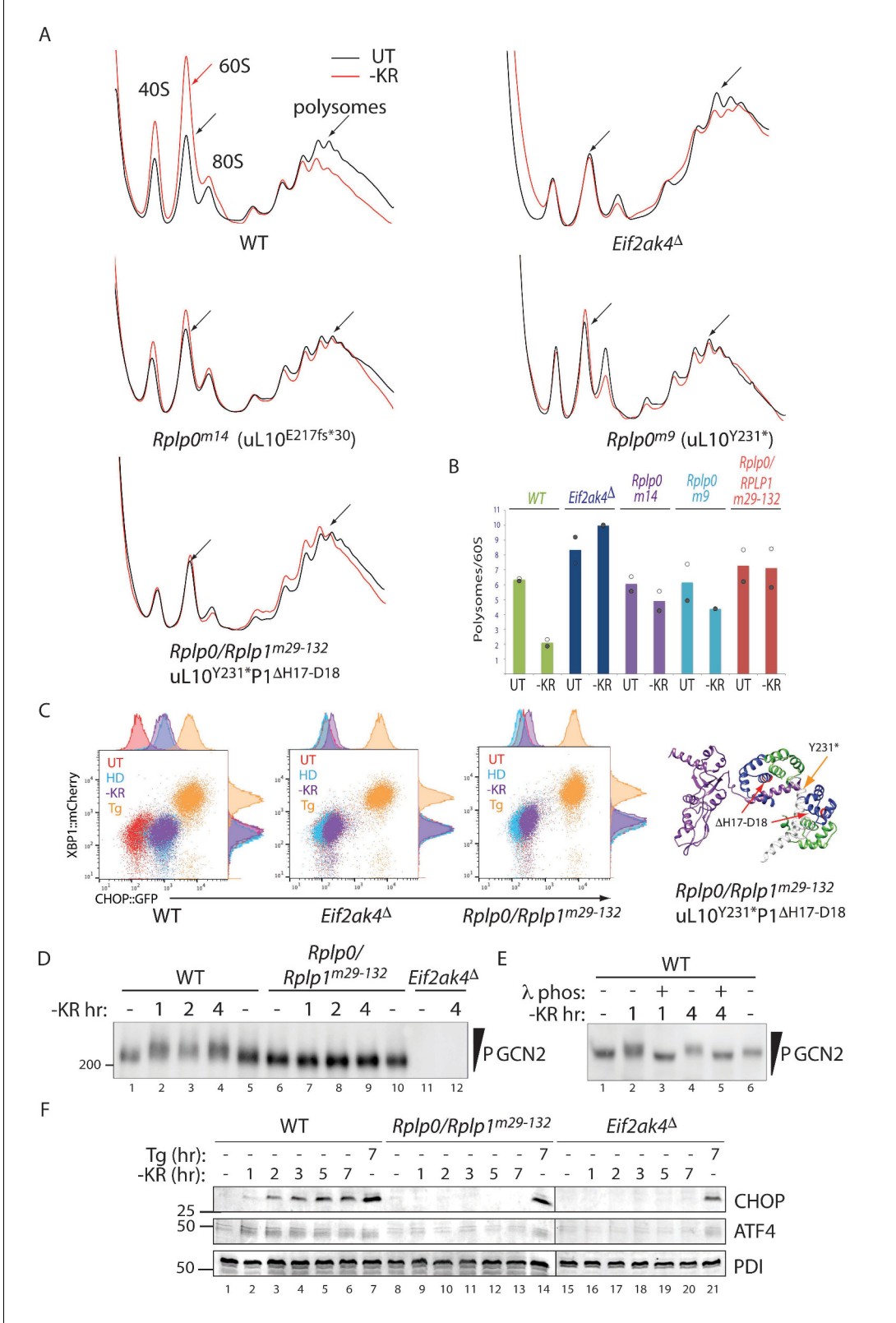

**Figure 3.** Impaired translational control, GCN2 activation, and ISR induction in amino acid starved P-stalk mutant cells. (**A**) Overlaid $A_{260}$ traces from 10–50% sucrose gradients loaded with cytoplasmic extracts from wildtype and the indicated mutant cells left untreated (UT, black) or starved for lysine and arginine (-KR, red). The 40S, 60S, 80S, and polysome peaks are labeled for orientation. Note the marked re-distribution of ribosomes from the denser polysome fractions to the lighter fractions in stressed wildtype cells, the lack of such redistribution in both the GCN2-ablated *Eif2ak4Δ* and

*Figure 3 continued on next page*

Figure 3 continued

compound P-stalk lesioned *Rplp0/Rplp1*$^{m29-132}$ mutants and the partial defect in the weaker *Rplp0*$^{m14}$ and *Rplp0*$^{m9}$ mutants. Shown are representative traces of A$_{260}$ (vertical axis) *vs.* sedimentation velocity (horizontal axis) from one of two such experiments. (B) Bar diagram of the ratio of the polysome to the free 60S subunit signal in stressed and unstressed cells of all five genotypes in the two experiments. The height of the bar reflects the mean of two experiments with values from matched experiments shown in open and closed circles respectively. (C) Flow cytometry analysis of wildtype (WT), GCN2-ablated (*Eif2ak4*$^Δ$) or the compound P-stalk *Rplp0/Rplp1*$^{m29-132}$ double mutant cells with treatments as in *Figure 2A* indicated. To the right is a ribbon diagram of the P-stalk (color coded as in *Figure 2B*). The red arrows point to the (H17–D18) deletion in P1 and the orange arrow to the truncation in uL10 at Y231. (D) GCN2 immunoblot of extracts from wildtype (WT), compound P-stalk *Rplp0/Rplp1*$^{m29-132}$ double mutant cells, or GCN2-ablated *Eif2ak4*$^Δ$ mutant cells, untreated or starved of lysine and arginine (-KR) resolved by phos-tag SDS-PAGE. (E) As in 'D' but following in vitro exposure of the lysate to λ phosphatase (λ phos). (F) CHOP and ATF4 immunoblots of extracts from wildtype (WT), compound P-stalk *Rplp0/Rplp1*$^{m29-132}$ double mutant cells and GCN2-ablated *Eif2ak4*$^Δ$ mutant cells, untreated, starved of lysine and arginine (-KR) or exposed to 1 μM thapsigargin (Tg) for the indicated time. Protein disulfide isomerase (PDI) serves as a loading control.

The online version of this article includes the following figure supplement(s) for figure 3:

**Figure supplement 1.** Impaired translational control, GCN2 activation, and ISR induction in amino acid starved P-stalk mutant cells (reporting on reproducibility of the observations shown in *Figure 3*).

apparent in the *Rplp0*$^{m14}$ single mutant (*Figure 4B*, middle panel), consistent with a destabilizing effect of the mutations on the association of P1 and P2 with the ribosome.

## Ribosomes isolated from P-stalk mutant cells are defective in stimulating GCN2 dependent eIF2α phosphorylation in vitro

While these studies were ongoing, we learned of findings, now published, indicating a physical interaction between GCN2 and the mammalian ribosomal P-stalk and providing evidence that intact ribosomes, or their isolated P-stalk, can stimulate GCN2-mediated phosphorylation of eIF2α in vitro (*Inglis et al., 2019*). At the concentrations used, isolated ribosomes had negligible associated eIF2α kinase activity. However, in our hands too, nanomolar concentration of ribosomes isolated from wildtype CHO cells markedly stimulated eIF2α phosphorylation by GCN2 (*Figure 5A* and *Figure 5—figure supplement 1A*). Interestingly, ribosomes isolated from the *Rplp0* and compound *Rplp0/Rplp1* mutant cells were impaired in GCN2-dependent eIF2α phosphorylation. This was evident in multiple preparations of wildtype and mutant ribosomes (*Figure 5A–5C* and *Figure 5—figure supplement 1A-C*). Furthermore, the hierarchy of the defect in GCN2 activation by the different ribosome preparations in vitro correlated with the ISR impairment of their source mutant cells in vivo, in that the in vitro defect was more severe in preparations of ribosomes from compound *Rplp0/Rplp1*$^{m29-132}$ mutant cells than its *Rplp0*$^{m9}$ single mutant parent or the unrelated *Rplp0*$^{m14}$ single mutant (*Figure 5A–5C* and *Figure 5—figure supplement 1C*).

The stimulatory effect of ribosomes appeared to be GCN2 specific, as the related PERK kinase was only minimally activated in vitro (*Figure 6A*). This finding is consistent with the lack of effect of the P-stalk lesions on the PERK-dependent ISR induction by ER stress in cells (*Figures 2A* and *3B–3F*).

To further characterize ribosome P stalk-dependent GCN2 activation, we compared experimentally accessible enzymatic features of GCN2 alone with those of a compound enzyme constituted of GCN2 and ribosomes. At physiological substrate concentrations of 2 μM eIF2α (*Chen et al., 2015*), the approximately 10-fold increase in the rate of GCN2-dependent eIF2α phosphorylation brought about by the presence of wildtype ribosomes (*Figure 5C*) could be mimicked by a 10-fold increase in the concentration of GCN2 in in vitro reactions without ribosomes. However, the relationship between reaction velocity and substrate concentration proved very different when comparing GCN2 (2.5 nM) with ribosomes (50 nM) to GCN2 alone (at 25 nM). While the former reaction saturates at less than 50 μM (substrate $K_m$ of ~7.8 μM (95% confidence 5.2–10.4 μM) and a $V_{max}$ of 130 min$^{-1}$ (95% confidence 116–140 min$^{-1}$)) the latter reaction was not saturated with substrate concentrations attainable experimentally (*Figure 6B and C*). The kinetic properties of GCN2, reacted with ribosomes from the mutant *Rplp0/Rplp1*$^{m29-132}$ cells, resembled GCN2 alone, establishing a role for the P-stalk in this shift in the enzymatic properties of the kinase (*Figure 6—figure supplement 1A and B*).

In keeping with recent observation (*Inglis, 2018* and *Inglis et al., 2019*), we also noted that ribosome-mediated potentiation of GCN2's eIF2α-directed kinase activity was not associated with a

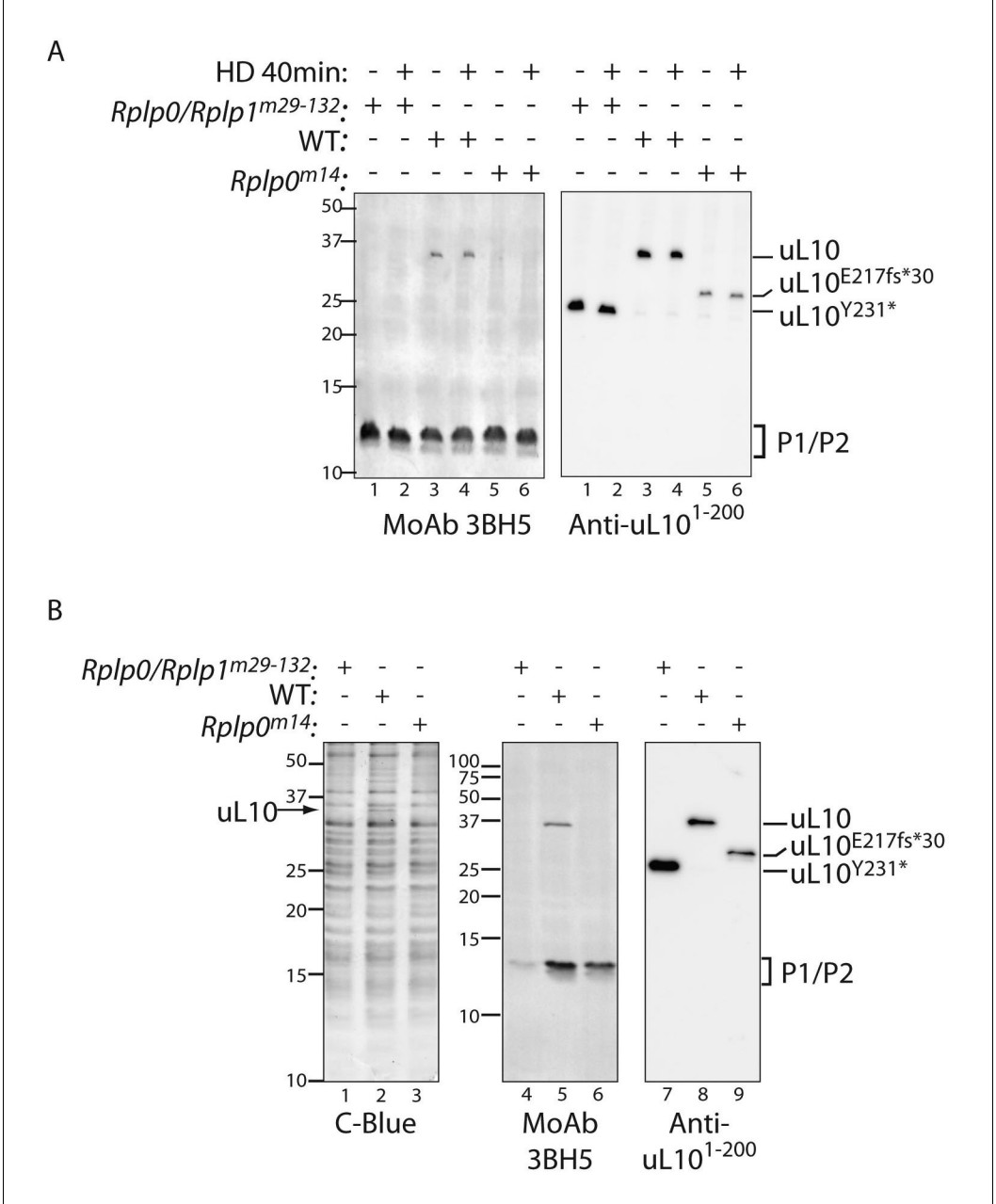

**Figure 4.** Defective ribosome association of P1 and P2 in mutant CHO cells. (**A**) Immunoblot of cytoplasmic extracts from untreated or histidinol-treated (HD, 0.5 mM) wildtype and the indicated mutant CHO cells. The polyclonal serum (anti uL10$^{1\text{-}200}$) recognizes the N-terminal portion of uL10 and the monoclonal antibody (MoAb 3BH5) recognizes the conserved acidic C-terminus of all three P-stalk proteins. The deletion mutants of uL10 are readily distinguishable in both immunoblots, but wildtype P1, P2, and the P1$^{\Delta H17\text{-}D18}$ mutant are not resolved. (**B**) Coomassie blue stained SDS-PAGE gel (left) and immunoblots of the protein content of ribosomes purified from wildtype (WT) or mutant CHO cells as in 'A' above. Note the near absence of P1/P2 proteins associated with ribosomes purified from the *Rplp0/Rplp1$^{m29-132}$* mutant cells, despite their presence in the whole cell extract (above).

consistent difference in the known GCN2 auto-activation mark, phosphorylation of activation-loop residue Thr899 (*Romano et al., 1998*) (*Figure 6D*). It is noteworthy that in vitro, ribosomes promote GCN2 phosphorylation on residues other than threonine 899 (*Inglis, 2018*). Unfortunately the antiserum directed towards human GCN2 pThr899 used here fails to recognise the hamster protein, thus

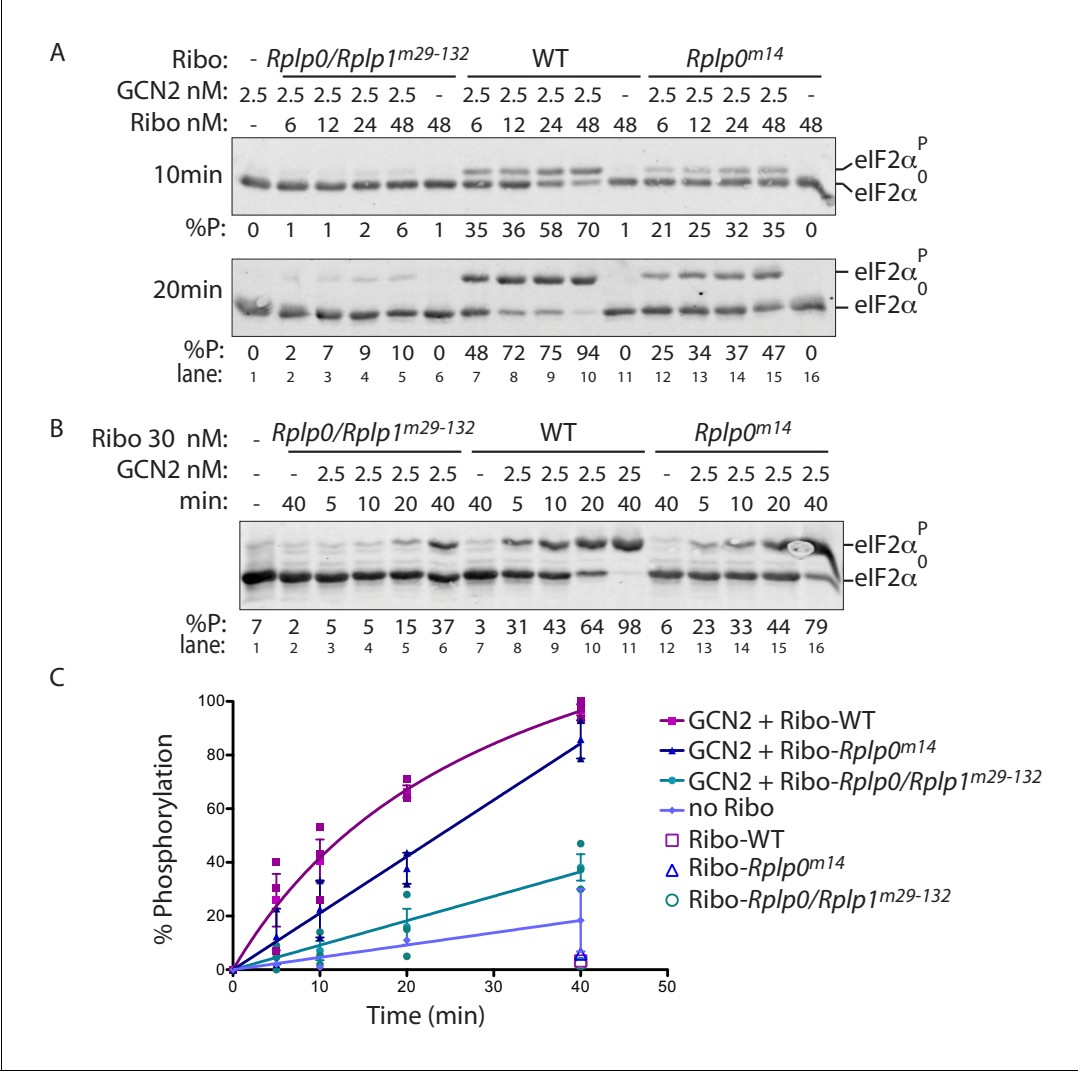

**Figure 5.** P-stalk lesions impair ribosome stimulation of eIF2α directed GCN2 kinase activity. (**A**) Immunoblot of eIF2α from in vitro phosphorylation reactions resolved by phos-tag SDS-PAGE. The concentration of purified GCN2 kinase, ribosomes, and the genotype of the ribosomes is indicated above the panel. The fraction of phosphorylated eIF2α is below the panel, the incubation time is on the left, and the migration of the phosphorylated (eIF2αP) and non-phosphorylated protein (eIF2α0) is on the right. (**B**) A time course of eIF2α phosphorylation reactions with 30 nM ribosomes annotated as in 'A' above. (**C**) Graph depicting the percent phosphorylation plotted against time from reactions shown in 'B' above and two similar experiments, all data points are shown along with the mean and range. (Shown are representative of experiments reproduced more than three times with independently isolated wildtype and mutant ribosomes).

The online version of this article includes the following figure supplement(s) for figure 5:

**Figure supplement 1.** P-stalk lesions (reproducibly) impair ribosome stimulation of eIF2α directed GCN2 kinase activity.

we are unable to ascertain if the defect in GCN2 phosphorylation observed in amino acid deprived, P-stalk mutant cells (*Figure 3D and E* and *Figure 3—figure supplement 1*) encompasses that residue. However, given that the phos-tag gels report on multiple phosphorylation events in the amino acid deprived wildtype cells that are absent from P-stalk mutant cells, it seems reasonable to conclude that the ribosome-mediated signal also results in phosphorylating events other than pThr899 in vivo. Their relationship to GCN2's activity as an eIF2α kinase remains to be determined. Together with the altered kinetics of the ribosome-associated enzyme, these features suggest a ribosome-dependent process extending beyond activation-loop autophosphorylation.

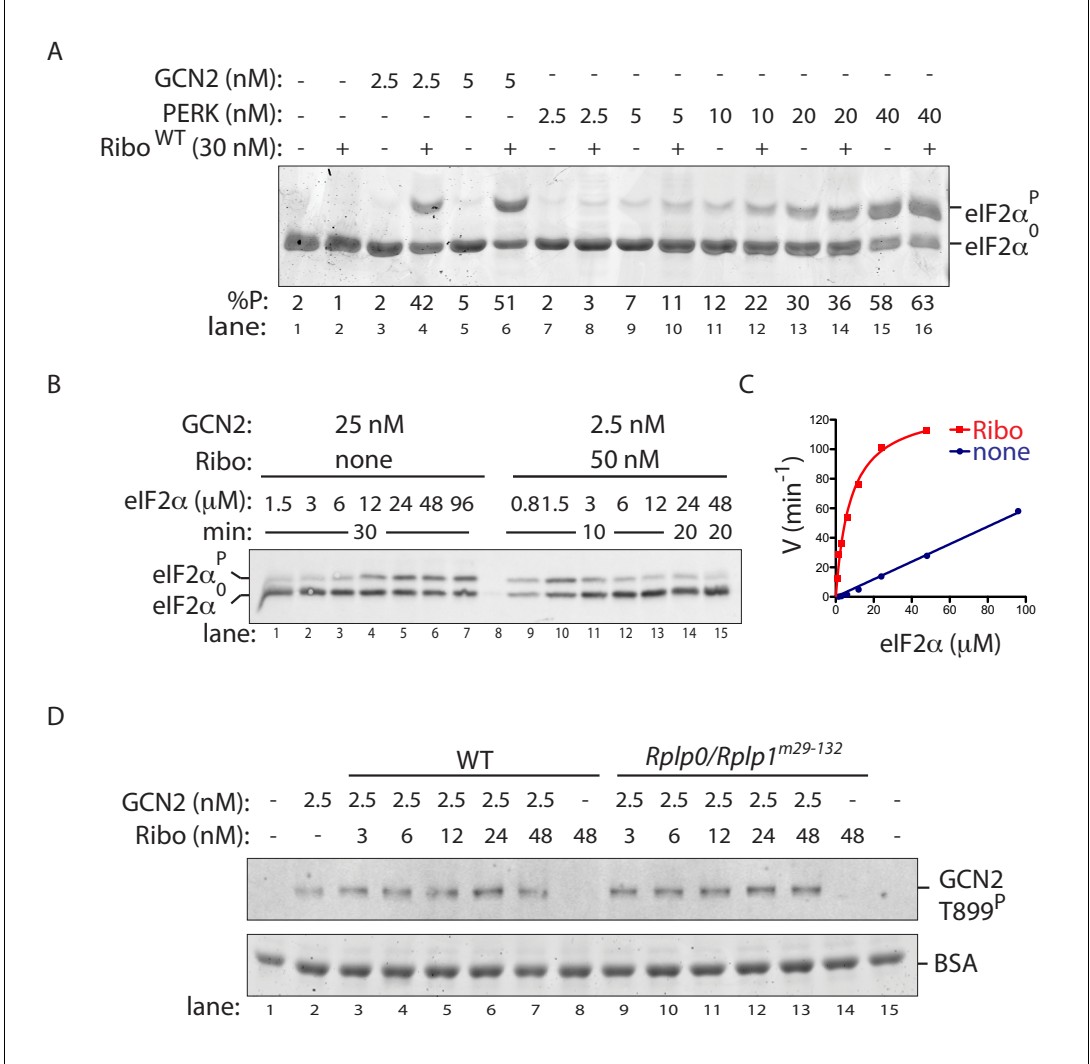

**Figure 6.** Ribosome activation alters GCN2 enzyme kinetics. (**A**) Coomassie-stained phos-tag PAGE of eIF2α from in vitro phosphorylation reactions with purified GCN2 or the cytosolic kinase domain of PERK, in presence or absence of wildtype ribosomes. (**B**) Immunoblot of eIF2α from in vitro phosphorylation reactions with escalating concentration of eIF2α substrate. Aliquots of each reaction containing equal amounts of eIF2α (~120 ng) were applied to the phos-tag gel. (Shown is a representative of experiments reproduced twice). (**C**) Plot of individual values of enzyme velocity (in $min^{-1}$) against substrate concentration (in μM) of the reactions shown in 'B' and Michaelis–Menten curve fit. Reactions with ribosomes appears to reach saturation at around 50 μM eIF2α whereas reactions without ribosomes exhibited no saturation at the highest concentrations tested. (**D**) <u>Top panel</u>: Immunoblot of phospho-GCN2-T899[P] signal from reactions incubated for 10 min with the indicated concentration of wildtype or mutant ribosomes in the presence of ATP. <u>Lower panel</u>: Coomassie-stain of the bovine serum albumin present in all reactions and serving as a recovery marker.

The online version of this article includes the following figure supplement(s) for figure 6:

**Figure supplement 1.** Ribosome activation alters GCN2 enzyme kinetics.

## Discussion

The correlation, established here, between a selective defect in GCN2-mediated ISR activation in cells with genetic lesions to their P-stalk and a defect in ability of their ribosomes to activate GCN2 in vitro implicates the P-stalk in propagating a signal from ribosomes perturbed by amino acid starvation to GCN2. This finding bridges recent genetic and cell biological observations implicating stalled ribosomes in GCN2 activation (*Ishimura et al., 2016*; *Darnell et al., 2018*) with biochemical evidence that the ribosome P-stalk can activate GCN2 in vitro (*Jiménez-Díaz et al., 2013*; *Inglis et al., 2019*), establishing that the in vitro observation, suggestive of a role for the ribosome

in GCN2 activation, relate to a process required for GCN2 activation by amino acid starvation in cells.

The implication of the P-stalk as an agent in GCN2 activation brings up interesting questions relating to the coupling between a process common to amino acid starvation and inhibition of tRNA charging and alteration in the state of the P-stalk. The location of the P-stalk on the surface of the ribosome adjacent to the A site and its functional role in recruiting elongation factors to the ribosome (*Helgstrand et al., 2007*; *Nomura et al., 2012*) and in stimulating their GTPase activity (*Mohr et al., 2002*), implicate the P-stalk in translation elongation. In elongating ribosomes, charged cognate tRNAs (in complex with eEF1A) followed by eEF2, cycle through the A site (reviewed in *Brown and Shao, 2018*; *Dever et al., 2018*). It is tempting to speculate on a scenario whereby the cycling of these factors in proximity to the P-stalk restrains the latter's GCN2 stimulatory activity. This may occur through elongation factor mediated steric blocking of the interaction between GCN2 and domain II of uL10 (shown to be essential for GCN2-P-Stalk binding, *Inglis et al., 2019*), by competition for the attention of the acidic C-terminal tails of the P-stalk proteins (which are important for activation of GCN2, *Inglis et al., 2019*), or by eliciting conformational changes in the P-stalk. Disruption of this elongation cycle by lack of cognate charged tRNA presumably stalls ribosomes in a conformation that relieves such restraint(s), exposing the dormant capacity of the P-stalk to activate GCN2 (*Figure 7*).

A ribosome-centered activation event, mediated by the P-stalk, fits the genetic data whereby in neurons lacking an abundant isoacceptor arginine tRNA, GCN2 activation is favored by lesions in GTPBP2-a mammalian ribosome rescue factor whose absence stabilizes stalled ribosomes (*Ishimura et al., 2014*; *Ishimura et al., 2016*). The notion that the activating signal arises from stalled ribosomes is also in keeping with the positive correlation noted between GCN2's response to single amino acid starvation in different mammalian cell lines and the extent of ribosome pausing (*Darnell et al., 2018*). Indeed, the relationship between stalling and GCN2 activity is likely homeostatic as GCN2$^\Delta$ cells exhibit more stalling in response to amino acid starvation (*Darnell et al., 2018*).

Ribosomes isolated from actively-translating fed cells (with repressed GCN2), were nonetheless potent activators of GCN2 in vitro (our observation and *Inglis et al., 2019*). Presumably ribosome isolation disrupts the aforementioned constraints on the P-stalk and exposes its latent ability to activate the kinase in vitro. In this vein, it is interesting to compare the P-stalk's role in yeast and mammalian GCN2 activation. Activation by the mammalian P-stalk requires both the ribosome-associated N-terminal domain II of uL10 and the acidic C-termini of the three P-stalk components that extend from the ribosomes surface (*Inglis et al., 2019*). By contrast, the yeast P1/P2 proteins are sufficient for GCN2 activation in vitro (*Jiménez-Díaz et al., 2013*). These observations fit the genetic evidence of a role for the free (not ribosome-associated) pool of yeast P2 in GCN2 activation in vivo. Interestingly, the free pool of yeast P2 contributes to GCN2 activation in response to osmotic stress and glucose deprivation, but is dispensable for GCN2 activation by histidine depletion. The latter observation is consistent with the notion that GCN2's response to amino acid starvation might depend on a ribosome-coupled P-stalk mediated signal, which may be provided adequately by yeast uL10 even in the absence of associated P1/P2 (*Jiménez-Díaz et al., 2013*).

The importance of the physical coupling between the P-stalk and the ribosome to mammalian GCN2 activation is suggested by the strong phenotype of the compound *Rplp0/Rplp1$^{m29-132}$* mutant in which the truncation of uL10 and the internal deletion of P1 conspire to deplete ribosomes of associated P-stalk acidic C-termini, whilst retaining a substantial pool of cytosolic P proteins with intact C-termini. The notion that efficient delivery of the P-stalk's message to GCN2 relies on proximity to the ribosome is consistent with the importance of other contacts made between GCN2 and the mammalian ribosome, such as those dependent on eS10 (*Lee et al., 2015*) (a product of a gene also 'hit' in our screen for CHOP::GFP dull cells). Whilst the ribosome takes central stage in GCN2 activation by amino acid starvation, it is interesting to consider that in mammals too there may be a role for the free pool of P1/P2 proteins in GCN2 activation in response to other non-ribosomal stress signals.

The nature of the P-stalk dependent activating event raises other questions. Yeast P1/P2 proteins markedly stimulated GCN2 autophosphorylation in vitro (*Jiménez-Díaz et al., 2013*). However, mammalian ribosomes had no consistent effect on the phosphorylation status of GCN2 Thr899 (here, *Inglis, 2018* and *Inglis et al., 2019*). Nonetheless, mammalian ribosomes imparted on GCN2

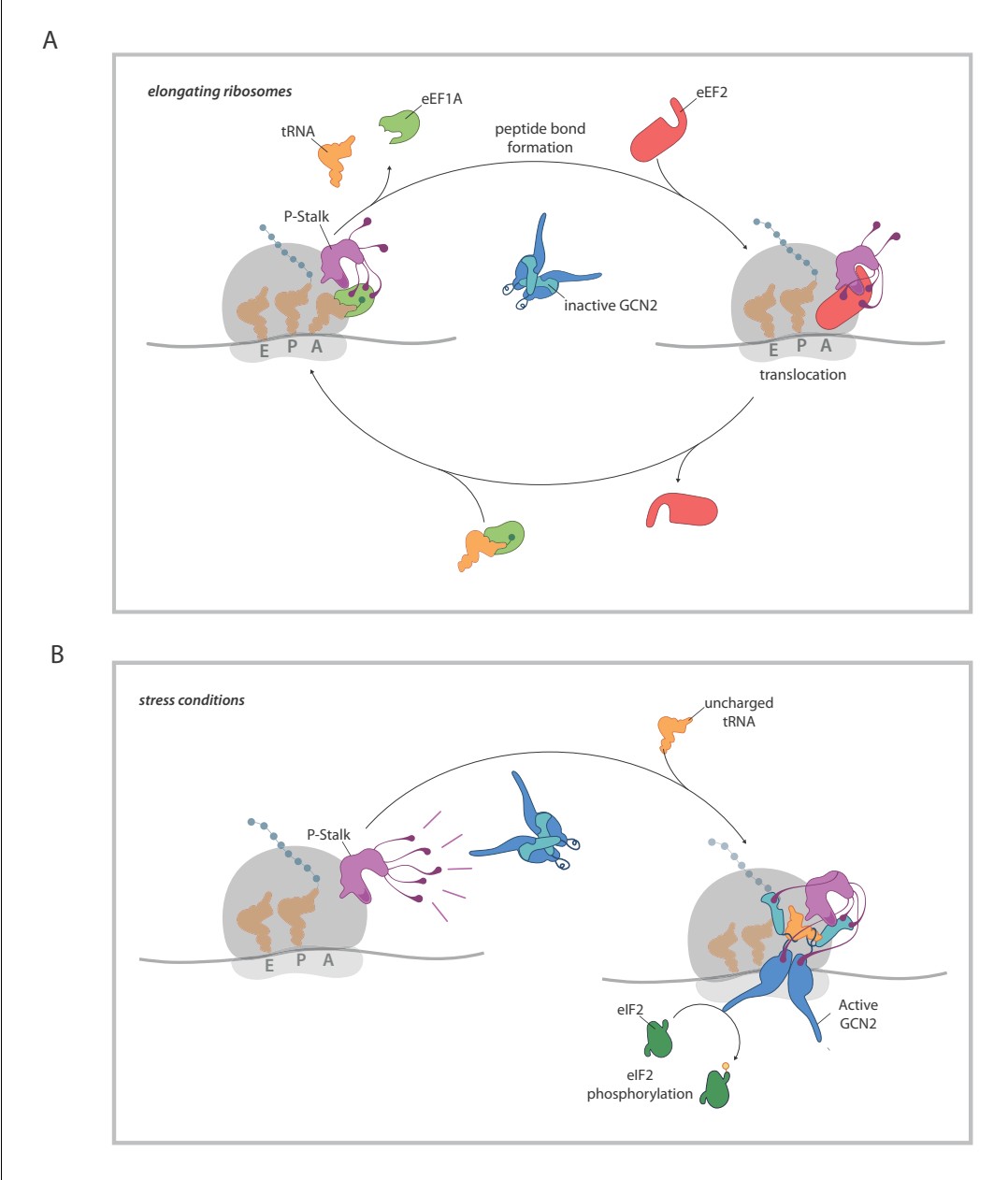

**Figure 7.** A model for ribosome state-dependent GCN2 activation by the P-stalk. (**A**) In amino acid replete cells, charged tRNAs and elongation factors eEF1A and eEF2 cycle through the GTPase associated centre repressing the P-stalk's latent ability to activate GCN2. Repression may arise from competition for domain II of uL10 (a GCN2 binding site, shaded deep purple), engagement of the acid C-terminal tails of the pentameric P-stalk (cartooned as tentacle-like extensions) or imposition of a conformation that is not conducive to GCN2 activation, amongst other mechanisms. (**B**) By limiting the pool of charged tRNAs, amino acid starvation or inhibition of tRNA synthetases disrupts the cycling of factors through the GTPase associated centre exposing the latent ability of the P-stalk to activate GCN2. This is speculatively presented as arising from the sequential binding of inactive GCN2 to the P-stalk and a subsequent activation step in which the acid C-terminal tails play a role. Active GCN2 (possibly in complex with uncharged tRNAs engaging its HisRS domain (pale blue)) phosphorylates its substrate, eIF2α.

new enzymatic properties, reflected in the lowering of the enzymes $K_m$. The nature of this activating event(s) is an open question, as is the linked question whether stimulation of the eIF2α-directed kinase activity of GCN2 requires the continued presence of ribosomes, or if the activated state survives dissociation of GCN2 from ribosomes.

The importance of GCN2's relationship with ribosomes to activation of the ISR (known in yeast as the GCN4-dependent General Control gene expression program) was first recognized before

identification of eIF2α as GCN2's substrate (*Ramirez et al., 1991*; *Zhu and Wek, 1998*) and subsequently buttressed by identification of ribosome-associated factors such as GCN1 and GCN20 as important GCN2 co-factors (*Marton et al., 1997*). These seminal early findings, together with more recent work (of which this paper is part), point to a role for a stalled ribosome-initiated P-stalk-mediated activation signal. Moreover, these observations suggest the possibility that GCN2, the first eIF2α kinase, evolved initially as part of a simple feed-back loop attenuating translation initiation in response to ribosome stalling. The coupling of GCN2's activity to a gene expression program that acts physiologically to relieve some of the more common causes of stalling, may have emerged later and eventually evolved into the ISR we recognize today in multicellular organism.

## Materials and methods

### Plasmid construction and materials

Standard cloning techniques were used to create the recombinant DNA vectors listed in the Key Resources Table (supplementary Table 1). The table also lists the antibodies, cell lines, reagents, oligonucleotides and software used in this study.

### Cell line construction

HeLa cells were maintained in DMEM supplemented with 10% Fetalclone II serum (Hyclone), 0.5% β-mercaptoethanol, 1x MEM-non-essential-amino-acids, and 1X pen-strep. CHO cells were maintained in Hams' F12 supplemented with 10% Fetalclone II serum (Hyclone) and 1x pen-strep. For lysine and arginine starvation, cells were washed 3x in PBS+ Mg2+ Ca2+ incubated in Advanced DMEM/F-12 (12634010, Thermo-Fisher) with 10% dialyzed FCS for 2–4 hr before the start of the experiment. Treated samples were then washed 3x with PBS + Mg2+ Ca2+ and incubated in SILAC Advanced DMEM/F-12 Flex Media (A2494301, Thermo-Fisher) supplemented with 17.5 mM glucose, and 10% dialyzed FCS for the indicated period.

HeLa cells were stably transfected with a linearized CHOP::GFP vector (lab m31) and pRc/RSV (Invitrogen) 10:1 and selected for G418 resistance and ISR induction. The selected line was then stably transfected with a constitutive mCherry expression vector to yield (Hela CGC55). A previously-described CHOP::GFP-C30 cell line (*Novoa et al., 2001*) was transfected with plasmid UK1313 pCAX-F-XBP1ΔDBD-mCherry (MP5) 10:1 with pbabe-puro (Addgene 1764) and a puromycin-resistant clone with tunicamycin induced XBP1::mCherry expression identified and the puromycin resistance was removed by transient transfection with sgCRISPR-RNA-expression vectors targeting the puromycin N-acetyltransferase gene (UK1901+UK1902) to yield the DP19 clone used throughout this study. Cas9 expressing derivatives of the HeLa CGC55 and CHO DP19 reporter lines were made by infecting the cells with lentivirus prepared from packaging Lenti-Cas9-Blast (UK1674, Addgene 1000000049) by co-transfection with helpers pMD2.g (UK1700, Addgene 12259), psPAX2 (UK 1701, Addgene 12260), in HEK 293 T cells and selecting for blasticidin resistance (10 and 30 μg/ml for HeLa and CHO respectively).

### CRISPR screening

Using the pooled human GECKO 2.0 CRISPR library and following an established protocol (Addgene 1000000049, *Sanjana et al., 2014*) we infected HeLa Cas9 expressing reporter cells library lentivirus at a MOI <0.3 and selected for Puro resistance resulting in a final representation of >60 surviving cells/guide for each of library segments A and B. After a week of expansion, cells were treated in arginine and lysine free medium for 20 hr followed by collection in PBS-EDTA, washing in PBS + 0.2% BSA and $8.47*10^7$ cells sorted with the dullest ~3% ($2.6 \times 10^6$) collected on an Influx or Melody cell sorter (BD biosciences). An equal number of treated cells were passed without sorting as a control group. The collected cells were replated, expanded and aliquots frozen or passed for another round of treatment and sorting.

Genomic DNA was prepared from $3.6 \times 10^7$ cells from each round using the DNAzol method (*Chomczynski et al., 1997*) from the separately maintained A and B library segments. PCR inserts were prepared for NGS sequencing with two rounds of PCR with primers UK 1433 and UK 1434 for the first round and primers UK 1432 and one of the barcoded reverse primers (UK1418-UK1432) for the second round. After quantification the products were subjected to NGS sequencing using

custom primer UK1435 and the illumina indexing primer with single-end reads of 50 bp on a hiseq 4000. The sequences were processed and guide counts, gene rankings and statistics were generated using MAGECK software (*Li et al., 2014*). The top 1% (200 hits) were submitted to metascape (*Zhou et al., 2019*) for gene annotation analysis and genes in the top enriched ontology clusters were added to the list along with the next ~9% of genes selected in the screen. Guides to 1500 unrelated CHO genes and 200 non-targeting control guides were also included in the library (NCBI Geo database; accession numbers awaiting assignment). CHO homologues were identified from a gene list from the CHO-K1 reference genome (GCF_000223135.1). Guides based on improved efficiency rules (*Doench et al., 2014*) were designed against the selected genes and pooled single stranded oligonucleotides (see oligo UK2580) were synthesized as part of a full genome library made with two sets of arms for independent PCR amplification and cloning (Twist Biosciences, California USA). The 19305 guide library (CHO-mini-library) targeting 3222 CHO genes (six guides per gene) was PCR amplified from the pool in 8 rounds of PCR with primers UK1855 and UK1856), digested with Bbs1, and the 27 bp fragments purified by PAGE and cloned into a BbsI cut UK1789 pKLV-U6gRNA (BbsI)-PGKpuro2ABFP vector (Addgene 50946) as described (*Koike-Yusa et al., 2014*).

The resulting focused library was used to mutagenize the CHO-CHOP::GFP, XBP1::mCherry double reporter cells at a MOI of 0.1 and a representation of ~350 cells for each library guide in duplicate pools. In the first round, 4–5% of the dullest cells following 20 hr induction with histidinol were selected and corresponding unselected pools maintained. In the second round three levels of dull cells corresponding to the lowest ~4.6, 11, and 24% were collected from each duplicate along with pools of treated unsorted cells. The selected guides were PCR amplified from genomic DNA as above but using primers UK1758 and UK1434 for the first round and primers UK1432 and one of the barcoded reverse primers (UK1759-UK1776) for the second round of PCR. The PCR products were subjected to NGS sequencing and analysis by MAGECK software as above for the HeLa screen. The figures show an average of the duplicates for both the dullest and medium dull selected cells (n = 4) compared to the average for the duplicate unselected controls (n = 2). The CHO CRISPR screening data has been submitted to NCBI Geo database (study accession number GSE134917).

Individual CRISPR-Cas9 mutant CHO cell lines were made by co-transfecting the indicated sgRNA expression vector with the Cas9-Blast plasmid (UK1674) into parental DP19 (CHOP::GFP; XBP1::mCherry reporter cells), selecting for the puromycin resistance encoded by the sgRNA expression vector (6 µg/ml) for two days. Reporter induction was measured by flow cytometry of at least 10000 gated live singlets on a Becton Dickinson LSR Fortessa on 2–4 independent experiments for each condition and mutant line.

The cell lines used in this study were all made in our lab from HeLa (ATCC Cat# CRL-7924, RRID: CVCL_0058) or CHO.K1 (ATCC Cat# CRL-9618, RRID:CVCL_0214) cell lines obtained from and authenticated by the ATCC. The identity of both the human (HeLa) the non-human (CHO.K1) cell lines has been authenticated using the criteria of A. successful targeting of essential genes using species specific CRISPR whole genome library, and B. sequencing of the wildtype or mutant alleles of the genes studied that confirmed the sequence reported for the corresponding genome. The cell lines have tested negative for mycoplasma contamination using a commercial kit (MycoAlert (TM) Mycoplasma Detection Kit, Lonza). None of the cell lines is on the list of commonly misidentified cell lines maintained by the International Cell Line Authentication Committee.

## Cell extracts and immunoblotting

Cells were treated as indicated and lysates prepared and ATF4 and CHOP detected using rabbit antiserum as previously reported except that proteins were transferred to PVDF and IR-800 (LI-COR) conjugated secondary antisera were used (*Harding et al., 1999*; *Harding et al., 2000*). Anti P-protein immunoblots were probed with a mouse monoclonal 3BH5 (1:5) that recognizes the conserved acidic C-terminus of all three P-proteins (*Vilella et al., 1991*) followed by anti-mouse IR800 and detection by LI-COR scanning and then the blot was incubated with rabbit monoclonal anti uL10 1– 200 (AbCam, Cambridge, UK ab192866, 1:1000), followed by anti-rabbit HRP and detection by chemiluminescence.

Samples containing CHO cell extracts (mock or λ phosphatase treated where indicated) or human GCN2 were denatured and separated on 5% Tris-acetate phos-tag or 7% Tris-acetate gels respectively and transferred in bicine transfer buffer (*Cubillos-Rojas et al., 2012*) and probed with anti-

GCN2 (phospho T899) antibody (1/1000, AbCam, Cambridge, UK ab75836) or an antibody raised against the bacterially-expressed kinase domain of mouse GCN2 antibody (lab name NY168) mixed 5:1 with non-affinity purified P-GCN2 890–904 (phospho T899) (human numbering, but the peptide is identical in mouse and human) (lab name 6779) in the case of CHO cell extracts (*Harding et al., 2000*) used at 1:3000.

## Polysomes and ribosome isolation

For polysome analysis cell lysis and 10–50% sucrose gradients were carried out as described in *Johannes and Sarnow (1998)*. Ribosomes were purified as described (*Khatter et al., 2014*) using modified buffers as follows: Cells were lysed in (15 mM Tris, pH 7.5, 0.5% IGEPAL, 6 mM MgCl2, 150 mM NaCl, 1 mM DTT, and 10 µl/ml RNAsin Plus (Promega), and protease inhibitors (Sigma S8830)), cleared 4 times at 18000g × 5 min and layered on a 3 ml sucrose cushion buffer D-30% (50 mM HEPES pH 7.5, 2 mM Mg(OAc)2, 150 mM KOAc, 30% sucrose), and centrifuged for 660 min at 40000 rpm in a TLA110 rotor. The pellet was resuspended in wash buffer (50 mM HEPES pH 7.5, 4.4 mM Mg(OAc)$_2$, 35 mM KOAc, 314 mM KCL, 6% sucrose 0.5 mg/ml puromycin, 1.2 mM GTP and 5 µl/ml RNAsin Plus) and rotated for 30 min at 4 ˚C to release nascent peptides, cleared for 10 min at 18000 g and then the supernatant re-pelleted through a sucrose cushion as described above. The final pellet was resuspended in the buffer D made with 6.8% sucrose and 2 mM TCEP.

## In vitro analysis of GCN2 activity (eIF2α phosphorylation assays)

GCN2, expressed as a StrepII-tagged protein was purified from insect Sf9 cells as described (*Inglis et al., 2019*) and stored in small aliquots at −80 °C. Phosphorylation reactions were carried out in PCR tubes at a total volume of 20 µL in an assay buffer consisting of 20 mM HEPES (pH 7.4), 50 mM potassium acetate, 5 mM magnesium acetate, 2 mM ATP, 0.5 mM tris(2-carboxyethyl)phosphine (TCEP), 0.05 mg/mL bovine serum albumin. Ribosomes isolated from CHO cells (or an equal volume of ribosome resuspension buffer D, see above) were combined with GCN2 (to a concentration of 2.5 nM in the final assay) and allowed to equilibrate at 17 ˚C for 10 min. The reaction was initiated by adding the N-terminal domain of human eIF2α (residues 1–185) purified from bacteria (*Ito et al., 2004*) to a final concentration of 2 µM, and allowed to progress at 17 ˚C for the indicated time until terminated by adding 6.6 µL of 4% SDS, 200 mM dithiothreitol in a tris-glycine buffer.

Samples were resolved on a 50 µM phos-tag/100 µM manganese chloride (Apex Biotechnology, Hsinchu City, Taiwan cat # F4002) 15% SDS-PAGE. The gel was soaked for 20 min in 50 mM EDTA, 0.1% SDS in tris-glycine buffer to chelate the phos-tag reagent and transferred onto a PVDF membrane and immunoblotted with a primary rabbit serum directed to the N-terminus of eIF2a (residues 1–185) (lab name NY1308) and an IRDye fluorescently labeled secondary anti-rabbit IgG (LI-COR) (*Chen et al., 2015*). The fluorescence signals were detected with an Odyssey near-infrared imager (LI-COR) and quantified by ImageJ (NIH).

## Acknowledgements

We thank the CIMR flow cytometry core facility team (Reiner Schulte, Chiara Cossetti and Gabriela Grondys-Kotarba) for help with cell sorting, Miguel Remacha from Centro de Biologia Molecular Severo Ochoa, Madrid, Spain, for the gift of MoAb 3BH5 to the P-stalk, The Lehner lab at CIMR for advice on CRISPR-Cas9 screens, Cambridge CRUK for access to their deep sequencing resource, the Warren lab at CIMR for use of their gradient maker, our colleagues, Claudia Rato, Steffen Preissler, Luke Perera, and Lisa Neidhardt for comments on the manuscript and Claudia Flandoli for the illustrations. This work was supported by Wellcome Trust Principal Research Fellowship to DR (Wellcome 200848/Z/16/Z), a Wellcome Trust Strategic Award to the Cambridge Institute for Medical Research (Wellcome 100140) and Cancer Research UK program grant to RLW (C14801/A21211).

## Additional information

### Competing interests

David Ron: Reviewing editor, *eLife*. The other authors declare that no competing interests exist.

## Funding

| Funder | Grant reference number | Author |
|--------|------------------------|--------|
| Cancer Research UK | C14801/A21211 | Roger L Williams |
| Wellcome | Wellcome 100140 | David Ron |
| Wellcome | Wellcome 200848/Z/16/Z | David Ron |

The funders had no role in study design, data collection and interpretation, or the decision to submit the work for publication.

## Author contributions

Heather P Harding, Conceptualization, Formal analysis, Supervision, Investigation, Visualization, Project administration; Adriana Ordonez, Formal analysis, Investigation; Felicity Allen, Leopold Parts, Software, Methodology; Alison J Inglis, Resources, Methodology; Roger L Williams, Resources, Funding acquisition, Methodology; David Ron, Conceptualization, Supervision, Funding acquisition, Investigation, Project administration

## Author ORCIDs

Heather P Harding  https://orcid.org/0000-0002-7359-7974
Roger L Williams  http://orcid.org/0000-0001-7754-4207
David Ron  http://orcid.org/0000-0002-3014-5636

## Decision letter and Author response

Decision letter https://doi.org/10.7554/eLife.50149.sa1
Author response https://doi.org/10.7554/eLife.50149.sa2

# Additional files

## Supplementary files

• Supplementary file 1. Key resources table.

• Transparent reporting form

## Data availability

All data generated or analysed during this study are included in the manuscript, supporting files, or are submitted to public databases.

The following dataset was generated:

| Author(s) | Year | Dataset title | Dataset URL | Database and Identifier |
|-----------|------|---------------|-------------|--------------------------|
| Harding HP, Ordonez A, Allen F, Parts L, Ingles AJ, Williams RL, Ron D | 2019 | CHO library B GCN Screen | https://www.ncbi.nlm.nih.gov/geo/query/acc.cgi?acc=GSE134917 | NCBI Gene Expression Omnibus, GSE134917 |

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
