## [Decision Letter]

**Acceptance summary:**

A role for the ribosome in contributing to GCN2 activation and thereby to the cells' response to amino acid starvation was originally suggested years ago, by findings that yeast GCN2 physically interacts with ribosomes and that essential co-factors to GCN2 activation, such as GCN1 and GCN20 are tightly ribosome-bound. Earlier this year, the Williams lab provided strong evidence that GCN2 transiently associates with ribosomes in vitro via a region of the ribosome known as the P-stalk and that the acidic tails of the P-stalk proteins can serve as GCN2 activators. However, the relevance of these important findings to GCN2 activation in cells remained unproven. This paper bridges the gap between the in vitro observations and cells, by showing that CRISPR/Cas9 engineered truncations of the P-stalk that compromise the ability of purified ribosomes to activate GCN2 in vitro also compromise GCN2 activation in amino acid starved cells. The findings here, therefore establish a role for the ribosome in conveying an amino acid starvation signal to GCN2. Though the details of that signal remain to be worked out, this study helps the field by telling us where to look for it.

**Decision letter after peer review:**

Thank you for submitting your article "The ribosomal P-stalk couples amino acid starvation to GCN2 activation in mammalian cells" for consideration by *eLife*. Your article has been reviewed by three peer reviewers, and the evaluation has been overseen by a Reviewing Editor and James Manley as the Senior Editor. The following individuals involved in review of your submission have agreed to reveal their identity: Evelyn Sattlegger (Reviewer #1).

The reviewers have discussed the reviews with one another and the Reviewing Editor has drafted this decision to help you prepare a revised submission.

Summary:

The authors report on the link between the ribosomal P-stalk and GCN2 activation. They performed a CRISPR/Cas9 screen to discover modifiers of GCN2-mediated integrated stress response (ISR) activation. From the many targets, they chose to study the ribosomal P-stalk, uL10, and P1 proteins because of a prior paper that demonstrated their importance in GCN2 activation in vitro. Importantly, they show here the P-stalk significance for ISR and GCN2 stimulation in vivo.

Essential revisions:

Please find below the full reviews of the three reviewers. As you can see they agree that the data are interesting and important, but the major reservation of the reviewers is that the novelty issue was not clearly and adequately addressed in this paper. It is important to show that the ribosomal protein-dependent activation of GCN2 occurs in vivo (in cells). The Williams paper (PNAS 2019) reported that isolated ribosomes promoted GCN2 activation, but there is the possibility that this could be an in vitro artifact. You showed by performing an independent genetic screening that the Williams' data are correct and thus your findings bolster these data. It is also significant that the mutants, which do not activate GCN2 in vivo, do not do it in vitro, either. Although they do not show that the ribosomal proteins activate directly GCN2, the data of this paper and those of Williams demonstrate that ribosomal proteins play an essential role in GCN2 activation. We assume that the reason you chose to select from the many targets to concentrate your study on ul10, P1, and P2 is because of the Williams paper and because these proteins provide a coherent target.

Reviewer #1:

This is a nice piece of work, this reviewer only has a few comments.

Introduction section: ".… not require elevated levels of uncharged tRNAs". The experiments used cannot exclude the possibility that there are localised changes in distribution of uncharged tRNAs (which cannot be detected after cell lysis), thus the statement "does not require elevated levels of uncharged tRNAs" should be phrased more carefully.

Though unlikely, is it possible that the function of the ribosome and its P-stalk merely is to help GCN2 to be efficient in accessing its substrate eIF2alpha, at least under the in vitro conditions used (Figure 6D), instead of to per se activate GCN2? May be good to elaborate on this as a possibility or why this is not likely.

Figure 6C, was this done from one experiment only or from several independent ones?

Results section paragraph five. It would be valuable to disclose in the publication, or as supplementary data, the names of found ribosomal proteins and translation initiation factors.

Reviewer #2:

This study investigates the link between the ribosomal P-stalk and GCN2 activation. The authors undertake a CRISPR/Cas9 screen to identify modifiers of GCN2-mediated integrated stress response (ISR) activation. Among the hits identified, they focus on sgRNAs targeting the ribosomal P-stalk, uL10 and P1. They then take advantage of in vitro biochemical assays to link P-stalk function to the ISR and GCN2 stimulation. This is a fast-moving field and much of the in vitro novelty of this study however has been preceded by a recent PNAS publication by the Williams group (PMID 30804176) (which the authors acknowledge and cite), but which steals the "thunder" from the current study.

I don't understand the rationale for keying in on *Rplp0* and *Rplp1* from the screening data. Looking at Figure 1E, there are a lot of sgRNAs that are enriched in the HD-dull sorted cells. What criteria places *Rplp0* and *Rplp1* above any of these other ones. Without this explanation, it looks like *Rplp0* and *Rplp1* were simply cherry-picked. The authors make the statement in the figure legend that "only *Rplp0* guides that were not depleted from the unselected pool of transduced cells were enriched in the CHOP:GFP dull population." I don't think that this is accurate since on the Figure I am given, I see 2 enriched guides for *Rplp0* (purple circles), 2 depleted guides, and 2 non-depleted guides. Hence 50% of the *Rplp0* guides that were not depleted were enriched.

Technically, the screen could have been better performed. In the initial screen, only 60 cells/guide were used which is quite low and likely to affect guide representation (most screens aim for >400 cells/guide). As I understand the Materials and methods, the top 10% genes were selected to go forward in the CHO screen. To this was added 1500 unrelated CHO genes and 200 non-targeting. How did these unrelated and non-targeting guides perform in the CHO screen? Were they truly neutral or they showed a wide spread. The authors need to show these controls and explain their behaviour.

Figure 1—figure supplement 1B shows a plot of the mean log2 fold change from each of the 3222 genes in the CHO mini library. It might be my inexperience, but I have not seen screening data presented in this manner without any statistics. What is the variation on these numbers? Where is *Rplp0* on this plot since only 2/6 guides showed activity in the primary screen (Figure 1E).

Figure 5C, 6C should have error bars.

The authors indicate in the Results that they identify sgRNAs that target the mRNA cap binding complex, other translation initiation factors, and ribosomal proteins, consistent with similar studies previously carried out in yeast. Can they elaborate and comment on this – also specifically what was found and how does the activity of these targets impinge on the GCN2 response.

In my opinion, the authors have to clearly indicate what is novel here in their study compared to the Williams group.

Reviewer #3:

Background:

Ishimura et al., 2016 observed an induction of the integrated stress response by GCN2 in the absence of accumulated uncharged tRNA, a known activator of GCN2, in mice brain with stalled ribosomes. This novel result implicated ribosome stalling with the activation of GCN2.

Jimenez-Diaz et al., 2013 showed that free yeast P1 and P2 proteins, which also form part of the ribosome P-stalk complex with uL10, activate mouse GCN2 autophosphorylation and eIF2alpha substrate phosphorylation.

Recently, Inglis et al., 2019 mapped interaction of human GCN2 with the ribosome (purified from rabbit reticulocyte lysate) to the uL10 subunit of the P-stalk by hydrogen deuterium exchange mass-spectrometry (HDX-MS). They also demonstrated addition of human P-stalk (consisting of uL10, P1/P2) potently activated GCN2 substrate phosphorylation in vitro. However, in contrast to what was observed for the yeast system, Inglis et al. did not observe GCN2 activation when they added only the human P1/P2 proteins to Gcn2 (ie the presence of uL10 was required).

In the submitted manuscript, Harding et al. further investigate the mechanism of mammalian GCN2 activation using a CRISPR-Cas9 genetic screen. Consistent with prior art, they uncovered a role for uL10 and P1 proteins of the P-stalk in the activation of Gcn2 in cells and in vitro.

Figure 1/2

The authors performed a CRISPR-Cas9 knock out screen using HeLa and CHO cell lines with GFP tagged CHOP, an effector of the ISR, as reporter of ISR activation. Cells were subjected to chemically induced amino acid starvation via inhibition of protein synthesis by histidinol. Cells impaired for CHOP-GFP activation appearing dull were isolated and the gRNA sequenced to identify genes involved in activation of the ISR upon amino acid starvation.

The authors incorporated appropriate controls by selecting for cells that showed impaired CHOP-GFP activation in response to histidinol, but remained ISR competent via activation of PERK by tunicamycin induced ER stress, avoiding false positive dull cells whereby ISR is entirely abolished. This gave confidence that dull cells would identify genes involved specifically in starvation activated ISR involving GCN2.

Two genes were identified in this screen namely the ribosomal P-stalk proteins uL10 and P1. These two hits were validated by observation of impaired CHOP-GFP indication upon histidinol treatment or lysine/arginine starvation in CHO cells with CRISPR-Cas9 targeted deletion of portions of uL10 and P1 protein.

Figure 3:

The authors looked at the effect of uL10 and P1 truncations on ISR – in terms of shift of ribosomes from polyribosomes to free 40s and 60s subunits measured by the ratio of the two upon induction of ISR and induction of ATF4 and CHOP proteins. Upon lysine/arginine starvation, a large increase in ribosome free subunits and a small decrease in polysomes was observed for WT cells. These signals were markedly attenuated by knock out of GCN2 and to a slightly less degree by double mutations in P1 and uL10.

Comment: The authors should make note of the fact (and explain) that the profile of WT cells in panel C does not look like the profile for the WT cells in Figure 2A. Specifically, the red blue and purple populations are fully distinct in Figure 2A but purple and blue overlap almost perfectly in Figure 3C.

Figure 4:

uL10 and P1 mutants showed impaired association of P1/P2 proteins with the ribosome by observation of decreased detection of P1/P2 protein from purified ribosomes with truncated uL10 compared to WT ul10; the authors showed this decrease cannot be explained by change in overall P1/P2 and uL10 protein level in cytoplasmic extract.

Figure 5:

Ribosomes activate the ability of GCN2 to phosphorylate eIF2alpha. Importantly, the extent of eIF2lapha phosphorylation correlated with level of association of P1/P2 protein with ribosomes.

Comment: The authors should consider inclusion of a catalytically dead GCN2 mutant.

Figure 6:

Ribosome activation alters GCN2 enzyme kinetics. The authors show a striking shift in GCN2 enzyme kinetics in response to ribosome addition. Consistent with flow-cytometry results, ribosome does not appear to activate the isolated kinase domain of PERK, lacking its regulatory domain apparatus (panel A).

Comment: The authors should consider using a full-length protein control to make for a more fair comparison (I doubt that the kinase domain of Gcn2 would be responsive to ribosomes either). In this regard, PKR would be easy to produce.

Comment: Panel B doesn't make sense as presented. Gcn2 appears more active without ribosome and the quantification in panel C doesn't match the gel pattern in panel B.

In panel D, the authors claim that GCN2 autophosphorylation, which was previously shown to be required for GCN2 catalytic activity, is not affected by ribosome addition.

Comment: The western blots are not compelling. The authors should consider repeating the experiment with fully dephosphorylated Gcn2 to make a stronger case. It appears that the starting Gcn2 material is already phosphorylated to a significant degree

Figure 7:

In a model figure, the author place GCN2 activation in the broader context of ribosome translation.

Comment: The complicated shapes of the protein in the cartoon make the figure hard to navigate – there is too much distracting complexity for the simple proposed model.

---

## [Author Response]

Essential revisions:Please find below the full reviews of the three reviewers. As you can see they agree that the data are interesting and important, but the major reservation of the reviewers is that the novelty issue was not clearly and adequately addressed in this paper. It is important to show that the ribosomal protein-dependent activation of GCN2 occurs in vivo (in cells). The Williams paper (PNAS 2019) reported that isolated ribosomes promoted GCN2 activation, but there is the possibility that this could be an in vitro artifact. You showed by performing an independent genetic screening that the Williams' data are correct and thus your findings bolster these data. It is also significant that the mutants, which do not activate GCN2 in vivo, do not do it in vitro, either. Although they do not show that the ribosomal proteins activate directly GCN2, the data of this paper and those of Williams demonstrate that ribosomal proteins play an essential role in GCN2 activation. We assume that the reason you chose to select from the many targets to concentrate your study on ul10, P1, and P2 is because of the Williams paper and because these proteins provide a coherent target.Reviewer #1:This is a nice piece of work, this reviewer only has a few comments.Introduction section: ".… not require elevated levels of uncharged tRNAs". The experiments used cannot exclude the possibility that there are localised changes in distribution of uncharged tRNAs (which cannot be detected after cell lysis), thus the statement "does not require elevated levels of uncharged tRNAs" should be phrased more carefully.

*Though unlikely, is it possible that the function of the ribosome and its P-stalk merely is to help GCN2 to be efficient in accessing its substrate eIF2alpha, at least under the* in vitro *conditions used (Figure 6D), instead of to per se activate GCN2? May be good to elaborate on this as a possibility or why this is not likely.*

The revised manuscript has been edited in line with these critiques. The possibility that localised changes in tRNA charging contribute to GCN2 activation in a manner influenced by the P-stalk is now explicitly acknowledged (Introduction and Abstract).

New Figures 3D, E and Figure 3—figure supplement 1, argues that the ribosome, via its P-stalk has a role in GCN2 activation, however this process may co-exist with other effects, such as modified enzyme-substrate interaction.

Figure 6C, was this done from one experiment only or from several independent ones?

The new Figure 6—figure supplement 1 now shows two further renditions of this experiment, comparing GCN2 activated by wildtype with P-stalk mutant ribosomes.

Results section paragraph five. It would be valuable to disclose in the publication, or as supplementary data, the names of found ribosomal proteins and translation initiation factors.

The full data set for the CHO screen are available at NCBI geo study GSE134917 as noted in the Materials and methods section of the updated manuscript.

Reviewer #2:This study investigates the link between the ribosomal P-stalk and GCN2 activation. The authors undertake a CRISPR/Cas9 screen to identify modifiers of GCN2-mediated integrated stress response (ISR) activation. Among the hits identified, they focus on sgRNAs targeting the ribosomal P-stalk, uL10 and P1. They then take advantage of in vitro biochemical assays to link P-stalk function to the ISR and GCN2 stimulation. This is a fast-moving field and much of the in vitro novelty of this study however has been preceded by a recent PNAS publication by the Williams group (PMID 30804176) (which the authors acknowledge and cite), but which steals the "thunder" from the current study.

Ingles et al., (PMID 30804176) suggest an unanticipated mechanism for signalling from the ribosome to GCN2. This, as the reviewer notes is thunderous. The added value of this study arises from the fact that here we take things one critical step further by documenting that these in vitro findings reflect a process *required* for GCN2 activation by amino acid starvation in cells. In other words we provide much needed evidence that the *suggestion*, by Inglis et al., that the ribosome may be implicated in propagating an amino acid starvation-induced signal to activate GCN2, is indeed correct. Given that not all plausible suggestions (hypotheses) that are based on in vitro experiments correctly describe the workings of cells, proving that a hypothesis is (or isn’t) a good description of what goes on in the cell can be a valuable contribution; thunderous or otherwise.

This point has been clarified by changes to the first paragraph of the Discussion.

I don't understand the rationale for keying in on Rplp0 and Rplp1 from the screening data. Looking at Figure 1E, there are a lot of sgRNAs that are enriched in the HD-dull sorted cells. What criteria places Rplp0 and Rplp1 above any of these other ones. Without this explanation, it looks like Rplp0 and Rplp1 were simply cherry-picked.

As explained clearly in the revised manuscript our focus on the P-stalk arose not from blind subservience to strength-of-signal hierarchies (the stuff of machines), but rather from a biased and informed interpretation of the data (such as humans are capable of). From the statistically-significant genes we were impressed (subjectively) by those encoding a biochemically related cluster of proteins (the P-stalk). We were further impressed that the pattern of guides targeting uL10, an essential protein, suggested that those affecting the C-terminal extra-ribosomal segment of the gene (the part that recruits the other P-stalk proteins) were enriched in cells displaying the ISR-deficient phenotype, whereas those located in regions encoding the ribosome associated mass of the protein were depleted.

Whilst such reasoning may qualify as *simple cherry picking*, we believe that sharing the reasoning that informed our choice with our readers has value. Our deliberate choice in the matter is now rendered explicit.

It may also be worth noting that whilst the experiments that followed were heavily informed by the findings of Inglis et al. (published earlier this year), our screen was completed in June 2017 and our interest in the P-stalk arose independently of knowledge of those seminal findings.

Figure 5C, 6C should have error bars.

New Figure 5C quantifies results from three experiments and shows the data points, means and fitted curves.

To showcase the reproducibility of the effect of wildtype ribosomes on GCN2 enzyme kinetics, we include as a new supplement to Figure 6 two experiments comparing wildtype and *Rplp0/Rplp1^m29-132^*compound mutant ribosomes.

Reviewer #3:In the submitted manuscript, Harding et al. further investigate the mechanism of mammalian GCN2 activation using a CRISPR-Cas9 genetic screen. Consistent with prior art, they uncovered a role for uL10 and P1 proteins of the P-stalk in the activation of Gcn2 in cells and in vitro.

Whilst our findings are certainly in agreement with previous in vitro work suggesting a role for the P-stalk in activating GCN2, we submit that there is no prior art on the importance of this process to GCN2 activation in cells. In other words, we provide much needed evidence that the *suggestion*, by others, that the ribosome *may* be implicated in propagating an amino acid starvation-induced signal to activate GCN2, is indeed correct. Given that not all plausible suggestions for how things work in cells are correct, proving, for the first time, that one is (or isn’t) can be a valuable contribution.

Figure 3:The authors looked at the effect of uL10 and P1 truncations on ISR – in terms of shift of ribosomes from polyribosomes to free 40s and 60s subunits measured by the ratio of the two upon induction of ISR and induction of ATF4 and CHOP proteins. Upon lysine/arginine starvation, a large increase in ribosome free subunits and a small decrease in polysomes was observed for WT cells. These signals were markedly attenuated by knock out of GCN2 and to a slightly less degree by double mutations in P1 and uL10.Comment: The authors should make note of the fact (and explain) that the profile of WT cells in panel C does not look like the profile for the WT cells in Figure 2A. Specifically, the red blue and purple populations are fully distinct in Figure 2A but purple and blue overlap almost perfectly in Figure 3C.

The difference in histidinol-induced changes in reporter activity presented in Figure 2A and Figures 1B, 2C, and 3C reflect experimental variation in the response of cells on different days to different lots of histidinol, etc. Such variation is expected, does not undermine interpretation of the experiment and will not surprise our readers.

Figure 4:uL10 and P1 mutants showed impaired association of P1/P2 proteins with the ribosome by observation of decreased detection of P1/P2 protein from purified ribosomes with truncated uL10 compared to WT ul10; the authors showed this decrease cannot be explained by change in overall P1/P2 and uL10 protein level in cytoplasmic extract.Figure 5:Ribosomes activate the ability of GCN2 to phosphorylate eIF2alpha. Importantly, the extent of eIF2lapha phosphorylation correlated with level of association of P1/P2 protein with ribosomes.Comment: The authors should consider inclusion of a catalytically dead GCN2 mutant.

The reviewer’s implicit concern that other contaminating kinases may contribute to eIF2α phosphorylation is addressed by measuring the phosphorylation in samples that lack GCN2 altogether. Furthermore, Inglis et al. have previously documented that the eIF2α -directed kinase activity in the preparations of GCN2 used here were entirely dependent on integrity of the GCN2 protein (see Figure 1 therein).

Figure 6:Ribosome activation alters GCN2 enzyme kinetics. The authors show a striking shift in GCN2 enzyme kinetics in response to ribosome addition. Consistent with flow-cytometry results, ribosome does not appear to activate the isolated kinase domain of PERK, lacking its regulatory domain apparatus (panel A).Comment: The authors should consider using a full-length protein control to make for a more fair comparison (I doubt that the kinase domain of Gcn2 would be responsive to ribosomes either). In this regard, PKR would be easy to produce.

Full-length PERK is a membrane associated protein, its isolation for biochemical experiments in vitro is impractical. Furthermore, given the absence of ribosomes in the ER lumen, the domain of PERK that is likely to benefit from any interaction with ribosomes is its cytosolic kinase domain, the one tested here as a relevant counterpoint to GCN2.

Comment: Panel B doesn't make sense as presented. Gcn2 appears more active without ribosome and the quantification in panel C doesn't match the gel pattern in panel B.

To extract enzyme kinetics, this gel is loaded with equal mass of substrate in each well not the whole reaction to prevent overloading in the high substrate lanes (see Materials and methods). The enzyme velocity in the corresponding reaction is derived from the product of the fraction of the protein phosphorylated and the substrate concentration.

In panel D, the authors claim that GCN2 autophosphorylation, which was previously shown to be required for GCN2 catalytic activity, is not affected by ribosome addition.Comment: The western blots are not compelling. The authors should consider repeating the experiment with fully dephosphorylated Gcn2 to make a stronger case. It appears that the starting Gcn2 material is already phosphorylated to a significant degree

Whilst we recognize the potential utility of commencing the reaction with dephosphorylated GCN2, the need to re-purify the kinase away from the phosphatase, renders this an impractical experiment. Thus we have further qualified our conclusions from this experiment, which we now interpret in light of the analysis of the ribosome’s role in GCN2 activation/phosphorylation in cells (new Figure 3D and 3E, and figure supplement): “It is noteworthy that in vitro, ribosomes promote GCN2 phosphorylation on residues other than threonine 899 (Inglis, 2018). Unfortunately, the antiserum directed towards human GCN2 pThr899 used here fails to recognise the hamster protein, thus we are unable to ascertain if the defect in GCN2 phosphorylation observed in amino acid deprived, P-stalk mutant cells (Figure 3D and 3E and Figure 3—figure supplement 1) encompasses that residue. However, given that the phos-tag gels report on multiple phosphorylation events in the amino acid deprived wildtype cells that are absent from P-stalk mutant cells, it seems reasonable to conclude that the ribosome-mediated signal also results in other phosphorylating events in vivo. Their relationship to GCN2’s activity as an eIF2a kinase remains to be determined.”

Figure 7:In a model figure, the author place GCN2 activation in the broader context of ribosome translation.Comment: The complicated shapes of the protein in the cartoon make the figure hard to navigate – there is too much distracting complexity for the simple proposed model.

We have simplified the figure by removing the lower part of the bottom panel.